# The Involvement of LAG-3^positive^ Plasma Cells in the Development of Multiple Myeloma

**DOI:** 10.3390/ijms25010549

**Published:** 2023-12-31

**Authors:** Natalia Kreiniz, Nasren Eiza, Tamar Tadmor, Ilana Levy Yurkovski, Sarah Matarasso Greenfeld, Adi Sabag, Raeda Mubariki, Celia Suriu, Ekaterina Votinov, Elias Toubi, Zahava Vadasz

**Affiliations:** 1The Division of Hematology, Bnai Zion Medical Center, Sderot Eliyahu Golomb 47, Haifa 3339419, Israel; natalia.kreiniz@b-zion.org.il (N.K.); tamar.tadmor@b-zion.org.il (T.T.); ilana.levy@b-zion.org.il (I.L.Y.); sarahmatarasso@gmail.com (S.M.G.); 2The Ruth and Bruce Rappaport Faculty of Medicine, Technion, Efron St 1, Haifa 3525433, Israel; 3The Proteomic Unit, Bnai Zion Medical Center, Sderot Eliyahu Golomb 47, Haifa 3104802, Israel; dr.niss@gmail.com (N.E.); adishay.sabag@gmail.com (A.S.); primrosethg@gmail.com (R.M.); elias.toubi@gmail.com (E.T.); 4The Division of Hematology, Galilee Medical Center, Nahariya-Cabri 89, Nahariyya 221001, Israel; celias@gmc.gov.il; 5Azrieli Faculty of Medicine, Bar-Ilan University, Henrietta Szold St 8, Safed 1311502, Israel; 6The Division of Hematology, Kaplan Medical Center, Derech Pasternak 1, Rehovot 7610001, Israel; ekaterinavo@clalil.org.il

**Keywords:** multiple myeloma (MM), Lymphocyte-Activation Protein 3 (LAG-3), regulatory molecules, Cluster of Differentiation-8 (CD8), hematological malignancies, plasma cells, cell-sorting

## Abstract

The Lymphocyte-Activation Protein 3 (LAG-3) inhibitory receptor is expressed on regulatory plasma cells (PCs). Micro-environmental cells that express LAG-3 were found to be increased during the progression of smoldering multiple myeloma (SMM). To assess the possible role of LAG-3 expression on regulatory PCs in patients with plasma cell dyscrasia. Purified Cluster of Differentiation 138 (CD138^+)^ PCs from patients with premalignant conditions, active multiple myeloma (MM), and controls were analyzed for the expression of LAG-3 by flow cytometry. Autologous CD8^+^T cells were incubated with sorted LAG-3^pos^ or LAG-3^neg^ PCs for 24 h. The expression of granzyme (Grz) in CD8^+^T cells was assessed by flow cytometry. LAG-3 expression on PCs in active MM (newly diagnosed and relapse refractory MM) was significantly increased compared to monoclonal gammopathy of undetermined significance (MGUS)/ SMM. Grz expression was significantly decreased in CD8^+^T cells incubated with CD138^+^LAG-3^pos^ PCs, compared to CD138^+^LAG-3^neg^ PCs in patients with plasma cell dyscrasia, *n* = 31, *p* = 0.0041. LAG-3 expression on malignant PCs can be involved in the development of MM from MGUS by decreasing the expression of Grz in CD8^+^T cells.

## 1. Introduction

Multiple myeloma (MM) is a clonal plasma cell malignancy characterized by the proliferation of antibody-producing plasma cells (PCs) within the bone marrow (BM), and the secretion of monoclonal immunoglobulins into the peripheral blood and the urine [1]. The increased proliferation of malignant PCs is frequently followed by excessive infiltration into the whole skeleton, causing bone lesions and the evolvement of renal failure, anemia, and hypercalcemia. In most cases, MM is preceded by an asymptomatic pre-malignant stage termed monoclonal gammopathy of undetermined significance (MGUS). This may then move into a more advanced but asymptomatic stage, defined as smoldering multiple myeloma (SMM) [2].

In clinical practice, three models are used to predict the progression of MGUS and SMM to MM: the Mayo Clinic, the Spanish Study group, and the IWMG 20/2/20 risk stratification systems [3,4,5]. These models include parameters of higher disease burden, such as a higher monoclonal protein level, a higher free light chain (FLC) ratio, and a greater percentage of monoclonal PCs in bone marrow (BM). In addition, these models include some biological markers of progression, such as the non-IGG type of M-protein in the Mayo Clinic model, the recognition of atypical PCs by flow cytometry, and the presence of immunoparesis, according to the Spanish model.

The prognosis of MM is defined by scoring systems, such as the International Staging System (ISS) and Revised ISS (R-ISS), which include B2-microglobulin, albumin, abnormal LDH, and high-risk genetic alterations [6,7]. Biological markers of MM progression are extensively studied but still have not found their place in clinical scoring models. One of the most investigated biological markers of progression in MM is PDL-1, which is also proposed as a potential therapeutic target [8,9]. CD200 is another surface molecule expressed on malignant PCs that was shown to have a prognostic significance in MM [10].

New treatment strategies, including different combinations of monoclonal antibodies, proteasome inhibitors, and immunomodulatory drugs, significantly improved the prognosis and survival of MM patients [11,12,13]. However, despite the progress that has been made in recent years in the treatment of MM, the disease remains incurable, and there is an unmet need for new therapies. The discovery of new therapeutical options in MM is directly related to gaining a better understanding of disease pathogenesis. From this point of view, the role of the immune system and the BM microenvironment in the development of MM has recently been emphasized, especially as a potential target for therapy.

Historically, IL-6 was one of the first immune molecules explored in the pathogenesis of MM. This soluble factor is secreted by plasma and bone marrow stromal cells, promoting the perfect microenvironment for MM development [14,15]. Unfortunately, early clinical trials of anti-IL-6 treatment failed to achieve a clinical response [16]. Similar to IL-6, TNF-α is one of the central cytokines that drive the proliferation and maintenance of tumor cells in MM [17]. Despite its important role in pathogenesis, there is no evidence of the effectiveness of anti-TNF-α antibodies in MM treatment; however, both IL-6 and TNF-α are correlated with disease activity in MM [17,18,19]. Another cytokine that was shown to be implicated in MM development is VEGF. The expression of VEGF by myeloma cell lines and PCs isolated from patients was demonstrated by S. Kumar [20]. Several antiangiogenic agents showed effectiveness in the preclinical phase but failed to achieve clinical response [21].

The recent discovery of anti-PD1 and anti-CTL4 antibodies started a revolution in the field of oncology. Unfortunately, these antibodies did not show efficiency in phase 3 clinical trials in MM and demonstrated a high level of toxicity [22,23]. One of the potential causes is the senescence of cytotoxic CD8^+^T cells in MM, which lack costimulatory receptors, such as CD28. The importance of cytotoxic CD8^+^T cells in the pathogenesis of MM is supported by the clinical effectiveness of CART cells and bispecific antibodies in MM. The failure of anti-PD1 therapy in MM encourages interest in researching other inhibitory checkpoints in MM, including LAG-3, TIM3, TIGIT, and BTLA [24]. These inhibitory proteins could be potential targets for immunotherapy in MM.

Recent attention has been focused on the LAG-3 molecule. While most studies are focused on the LAG-3 role in the microenvironment, less is known about LAG-3 expression on malignant PCs and its possible role in the progression of MM from MGUS and SMM [25,26]. A growing interest in LAG-3 was influenced by the recent discovery of a new subgroup of PCs in mice, called regulatory PCs, which express LAG-3 and secrete IL-10 [27,28,29]. These cells have not been studied in humans, and there is no data about their role in the pathogenesis of MM. Therefore, the aim of our study was to assess the possible role of LAG-3 expression on regulatory PCs in patients with plasma cell dyscrasia. We hypothesize that malignant PCs expressing LAG-3 are actively involved in the pathogenesis of MM progression from MGUS and SMM by impairing the function of effector immune cells.

## 2. Results

### 2.1. Demographic and Clinical Characteristics of Patients with Plasma Cell Dyscrasia

Eighty-one patients (44 males and 36 females) with a mean age of 70 ± 9 years were enrolled in this study. The patients were assigned by senior hematologists based on International Myeloma Working Group (IMWG) criteria to four subgroups: MGUS (n = 17), SMM (n = 20), active MM (n = 27), and RRMM (n = 5). Patients who were found to have normal bone marrow biopsies were defined as controls (n = 12).

Thirteen of the patients (19%) had high-risk FISH results (del(17p), t(4:14), and t(14:16)). A total of 34 of the patients (50%) had monoclonal IgG serum levels, 12 patients (17%) had monoclonal IgA serum levels, 3 patients (4%) had monoclonal IgM serum levels, and 12 patients (17%) had abnormal free light chain (FLC) levels.

In addition, 34 patients (49%) had immunoparesis (defined as a decrease in one or more of the uninvolved immunoglobulins below the normal range). (Table 1).

### 2.2. CD138^+^ Plasma Cell Characterization

#### 2.2.1. The Expression of Regulatory Receptors on CD138^+^ Plasma Cells

In order to characterize regulatory PCs in active MM patients and premalignant conditions (MGUS and SMM), the expression of LAG-3, PDL1, CD200, and PD1 receptors on purified PCs was assessed using flow cytometry.

The mean expression of LAG-3 in active MM patients was significantly higher compared to the control group (27.69 ± 3.27% vs. 2.441 ± 0.51%, mean ± SEM, *p* < 0.0001, respectively), and was also higher compared to premalignant conditions (16.58 ± 2.85%, mean ± SEM, *p* = 0.025), as shown in Figure 1A.

In addition, the mean expression of PDL1 in active MM patients was also significantly higher compared to the control group (15.46 ± 6.65% vs. 2.448 ± 1.05%, mean ± SEM, *p* = 0.33, respectively), as shown in Figure 1B.

The mean expression of PD1 and CD200 was not found to be significantly different among the study groups, as seen in Appendix A.

#### 2.2.2. The Secretion of Tumor-Related Pro-Inflammatory Cytokines from CD138^+^ Plasma Cells

Pro-inflammatory cytokines IL-6 and TNF-α play essential roles in the pathogenesis of MM.

LAG-3^pos^ PCs were purified and sorted, as described in Section 4. While using multiplex analysis of cytokines, following 24 h of incubation, we found that LAG-3^pos^ PCs secreted higher levels of IL-6 compared to LAG-3^neg^ PCs (36.29 ± 15.38 pg/mL vs. 17.01 ± 6.31 pg/mL, mean ± SEM, respectively), as seen in Figure 2A. Moreover, when we analyzed the secretion of TNF-α in these cells, we could also demonstrate an increased level of secretion in LAG-3^pos^ PCs compared to LAG-3^neg^ PCs (16.18 ± 6.77 pg/mL vs. 6.91 ± 6.91 pg/mL, mean ± SEM, respectively), as seen in Figure 2B.

### 2.3. Correlation of LAG-3 Expression with Laboratory and Clinical Parameters

The expression of CD81 and CD56 on PCs using flow cytometry is considered routine immunophenotyping in the diagnosis and follow-up of patients with plasma cell dyscrasia.

These two markers were found to be correlated differently with LAG-3 expression on PCs from plasma cell dyscrasia patients. LAG-3 showed a positive correlation with CD56 (r = 0.680, *p* = 0.043) Figure 3A. However, CD81 only demonstrated a trend of negative correlation with LAG-3 (r = −0.59, *p* = 0.0973), as seen in Figure 3B.

Ferritin is a commonly used marker for iron storage assessment. In addition, ferritin is a marker of acute and chronic inflammation, which is associated with some cancers. High levels of ferritin have adverse prognostic significance in MM patients [30]. When we analyzed and correlated the serum ferritin level of patients with active MM, we found a positive correlation with LAG-3 expression (r = 0.59, *p* = 0.04): see Figure 3C.

### 2.4. The Biological Function of LAG-3^pos^ Plasma Cells

#### 2.4.1. Granzyme Secretion from CD8^+^T Cells Co-Cultured with Plasma Cells

In order to examine the biological function and significance of LAG-3 expression, we assessed the cytotoxicity potential of autologous CD8^+^T cells following co-culture with PCs by measuring granzyme (Grz) expression; CD8^+^T cells co-cultured with LAG-3^pos^ PCs showed a decrement in the expression of Grz compared to cells co-cultured with LAG-3^neg^ PCs (13.68 ± 5.382% vs. 6.426 ± 1.953%, respectively, mean ± SEM, *p* = 0.0041). This result might point toward a reduction in CD8^+^T cell cytotoxicity potential as an effect of LAG-3^pos^ PCs, as seen in Figure 4. Furthermore, in the sub-group analysis, this reduction in Grz expression on CD8^+^T cells was found to be more significant in patients with active MM.

#### 2.4.2. Cytokine Secretion Pattern of CD8^+^T Cells Cultured with Plasma Cells

In order to examine the secreted cytokines of CD8^+^T cells co-cultured with either LAG-3^pos^ or LAG-3^neg^ PCs, we analyzed the secretion of TNF-α and IL-6 from the supernatants. The reported cytokine levels reflect the combined activity of CD8^+^T and PCs since both cells secreted IL-6 and TNF-α differently.

We have found that CD8^+^T cells secrete high levels of IL-6 (49.04 ± 18.70 pg/mL mean ± SEM), and its secretion was reduced upon co-culturing with PCs. This reduction was more pronounced when CD8^+^T cells were co-cultured with LAG-3^pos^ PCs compared to LAG-3^neg^ PCs (13.58 ± 13.58 pg/mL vs. 31.56 ± 14.75 pg/mL, mean ± SEM, *p* = 0.09 respectively).

When we analyzed the secretion of TNF-α, another pro-inflammatory cytokine, we were able to demonstrate that CD8^+^T cells secrete a relatively high amount of TNF-α, and the addition of malignant LAG-3^pos^ PCs cells reduced its secretion level (18.46 ± 6.45 pg/mL vs. 7.64 ± 2.96 pg/mL, *p* = 0.1, respectively), while the TNF-α levels were increased when CD8^+^T cells co-cultured with LAG-3^neg^ PC, (39.24 ± 22.16 pg/mL): Figure 5.

#### 2.4.3. CD8^+^T Cells Effect on Plasma Cells

VEGF is one of the angiogenic factors responsible for increasing the formation of new vessels in the bone marrow and stimulating the secretion of IL-6, which contributes significantly to the pathogenesis of MM [20]. Thus, we were interested in exploring the secretion pattern of this angiogenetic factor. The multiplex analysis results showed that CD8^+^T cells alone do not secrete VEGF, while it is secreted equally from both LAG-3^pos^ and LAG-3^neg^ PCs alone (14.45 ± 8.68 pg/mL vs. 15.12 ± 8.07 pg/mL, mean ± SEM, respectively), as seen in Figure 6A. Surprisingly, when we co-cultured CD8^+^T cells with LAG-3^pos^ PCs, a reduction in the VEGF secretion level was shown, and the VEGF secretion level showed no effect following co-culture with LAG-3^neg^ PCs (9.7 ± 3.567 pg/mL vs. 19.67 ± 4.524 pg/mL, mean ± SEM, *p* = 0.015), as seen in Figure 6B.

## 3. Discussion

B regulatory cells are front players in regulatory immune responses, the impaired function of which is followed by the failure to maintain self-tolerance and the development of autoimmunity [30]. In addition to their role in autoimmunity, they allow malignant cells to survive through their increased expansion and production of IL-10 and IL-35 inhibitory cytokines, which are suppressed by anti-tumor immune responses [31].The recent recognition of natural regulatory PCs, characterized by the increased expression of inhibitory receptors LAG-3, CD200, PD-L1, and PD-L2, revealed that they are significant source of B cell-derived IL-10, allowing for the suppression of anti-tumor immune responses and tumor cell spread [32].

LAG-3 protein is encoded by the LAG-3 gene located on human chromosome 12 [33] and is expressed on the membranes of B, NK, and dendritic cells [34]. LAG-3^pos^ CD138^+^ regulatory PCs were shown to develop via antigen-specific mechanisms [32]. When stimulated via a Toll-like receptor-driven mechanism, they do not proliferate but they do increase their IL-10 expression and secretion [29]. Although it is poorly defined, the role of LAG-3 in hematological malignancies has been increasingly assessed. Therefore, LAG-3 was found to be expressed both on T and malignant B cells in diffuse large B cell lymphoma (DLBCL) and chronic lymphocytic leukemia (CLL), especially in Richter transformation (RT), and had a negative impact on progression-free survival (PFS) and overall survival (OS) in DLBC [35,36]. Increased micro-environmental cell LAG-3 expression in SMM contributed to the suppression of anti-tumor T-cell responses and the increased progression to symptomatic MM [26]. In the recent study by Jooeun Bae et al., treatment with anti-LAG-3 antibodies in vitro enhanced the anti-myeloma immune response by increasing the proliferation and functional capacity of cytotoxic T cells [25]. According to our knowledge, the current study is the first to show that LAG-3 expression is significantly increased on PCs obtained from the bone marrow of patients with symptomatic MM compared to MGUS/SMM and controls.

According to previous knowledge, PDL1 on plasma cells plays a role in the progression of MM from a premalignant state, and its high expression on plasma cells in active MM is associated with a poor prognosis [37]. For instance, Costa F et al. demonstrated higher expression of PDL1 in patients with MM than in healthy volunteers and MGUS, but not in SMM, proposing the possible benefit of therapeutic intervention by interrupting the PD1/PDL1 axis early at the MGUS stage [38]. Regarding its prognostic role in MM, PDL1 expression was associated with poor overall survival in active MM [8]. Our study showed a trend for higher PDL1 on PCs in active MM compared to MGUS/SMM, but this was not confirmed statistically. Regarding CD200, in the literature, this was steadily expressed on PCs in MM during the treatment process, making it a helpful follow-up marker [39]. In addition, it was found to have prognostic significance in MM patients [10,40].

Our finding of LAG-3 expression on malignant PCs strongly supports its possible role in predicting the progression of MGUS and SMM into symptomatic multiple myeloma. This conclusion is further supported by its positive correlation with CD56 in patients with PCD, and higher serum ferritin level in MM.

These results prompted us to explore the biological significance of LAG-3 expression on PCs in the development of MM. For this purpose, autologous CD8^+^T cells were co-cultured with LAG-3^pos^ vs. LAG-3^neg^ PCs purified from patients with MGUS, SMM, or active MM, and analyzed for Grz and Per expression. The supernatants of the co-cultures were evaluated for pro- and anti-inflammatory cytokines. The hypothesis was that LAG-3 on PCs suppresses the function of cytotoxic CD8^+^T cells by decreasing their expression of Grz, Per, and the secretion of pro-inflammatory cytokines. Cytotoxic CD8^+^T cells are important for immunosurveillance in cancer and in MM [41,42]. This statement is strongly supported by the clinical effectiveness of immunotherapy, designed to enhance the function of cytotoxic T cells either by blocking the checkpoint receptors on these cells or by constructing a chimeric antigen receptor T cell (CAR T) [43]. Cytotoxic T cell exhaustion and senescence were previously described in MM patients, especially at the tumor site [44], but the exact mechanism of CD8^+^T cell decline is not known. Our results from the experiment with co-cultures revealed that LAG-3 on PCs decreased the Grz expression in CD8^+^T cells in the entire study population, including patients with MGUS, SMM, and active MM. In the subgroup analysis, the decrease in Grz achieved statistical significance in patients with MM, but a clear tendency was also found in MGUS and SMM. This could be explained by the higher expression of LAG-3 on PCs in patients with active MM, caused by the effect of amplification. Next, we measured pro- and anti-inflammatory cytokines in the supernatants of CD138^+^LAG-3^pos^, CD138^+^LAG-3^neg^, CD8^+^T cells and co-cultures of CD8^+^T cells with LAG-3^pos^ or LAG-3^neg^ PCs.

The secretion of pro-inflammatory cytokines, such as TNF-α and IL-6, was higher in supernatants of LAG-3^pos^ PCs, but in co-cultures, the results were contradictory. The results did not achieve statistical significance, probably due to the small sample size, but there was a clear trend. According to previous knowledge of MM, the secretion of TNF-α and IL-6 stimulates the survival of malignant PCs [14,45]. We cannot know precisely what happens during the interaction between T and PCs in co-cultures, but based on our findings, we can assume that analogous to the expression of Grz in our previous experiment, LAG-3 decreases the secretion of TNF-α and IL-6 by CD8^+^ CTLs. The limitation of the co-culture studies is that they do not represent the entire bone marrow microenvironment. We can conclude that LAG-3^pos^ PCs possibly have a survival advantage over LAG-3^neg^ PCs by secreting more TNF-α, but also have an immunosuppressive effect on CD8^+^ CTLs, by decreasing the secretion of the same cytokines from CD8^+^ CTLs.

In our study, there was a significant decrease in VEGF secretion in co-cultures of LAG-3^pos^ PCs with CD8^+^ CTLs in patients with PCD. In MM, VEGF increases the formation of new vessels in the bone marrow and stimulates the secretion of IL-6, contributing significantly to the pathogenesis of the disease [20]. In addition, there is increasing evidence that VEGF plays different roles in immunology, both pro- and anti-inflammatory, but was shown to have primarily immunosuppressive effects in cancer [46,47]. Regarding the fact that VEGF was shown to have primarily immunosuppressive effects in cancer, we hypothesized that VEGF secretion would increase in co-cultures with LAG-3^pos^ PCs. However, the opposite results occurred: there was a significant decrease in VEGF secretion in co-cultures of LAG-3^pos^ PCs with CD8^+^ CTLs in patients with PCD. We are still unable to explain this phenomenon; however, one possibility is that more VEGF was bound to its receptor on CTLs in co-cultures with LAG-3^pos^ PCs than with LAG-3 PCs. More studies are warranted in order to better clarify this phenomenon. This study is the first to describe the significance of the expression of LAG3 on plasma cells in the evolution of MM. The limitation of the study is its relatively small cohort. Future studies are needed in order to establish these important findings.

## 4. Materials and Methods

### 4.1. Patients

Bone marrow aspirates and peripheral blood samples were obtained from patients from three medical centers in Israel—Bnai Zion, Kaplan, and Galilee Medical. All participants signed the informed consent form, and each hospital’s ethics committee approved the study. The study was conducted in accordance with the Declaration of Helsinki, and approved by the Institutional Review Board of Bnai Zion medical center (0015-19-BNZ, date: 4 February 2019).

The study participants were assessed by senior hematologists and, based on the International Myeloma Working Group (IMWG) criteria, assigned to four subgroups: MGUS, SMM, active MM, and RRMM. Patients in whom bone marrow aspirates and biopsies were found normal were defined as controls.

Laboratory and clinical data were collected from the electronic data system (data shown in Table 1). These include FLC level; monoclonal immunoglobulin serum level; β2-microglobulin level; mcg/L; MM stages according to R-ISS criteria; C reactive protein (CRP); ferritin; FISH; and flow-cytometry results (CD20, CD19, CD81, CD56, CD38).

### 4.2. Cell Isolation

A total of 8 ml of fresh bone marrow aspirates (BMA) and 40 mL of fresh peripheral blood (PB) were drawn to heparin-washed tubes. BMA or PB were loaded on Lymphoprep (07851, STEMCELL Technology, Vancouver, BC, Canada) and centrifuged at 800× *g* at room temperature continuously for 35 min. After centrifugation, the BM mononuclear cells (BMMCs) were collected, and PCs were purified using anti-human CD138 magnetic microbeads (#130-051-301, MACS Miltenyi Biotec, Bergisch Gladbach, Germany). In addition, CD8^+^T cells were isolated from PB mononuclear cells (PBMCs) using anti-human CD8 magnetic beads (#130-045-201, MACS Miltenyi Biotec, Bergisch Gladbach, Germany) according to the manufacturer’s instructions.

### 4.3. Flow Cytometry

Cells were stained with specific monoclonal antibodies against extracellular markers for 30 min at room temperature, followed by a wash. For intracellular staining, the procedure was conducted using FIX and PERM kit (GAS004, Invitrogen, Waltham, MA, USA) according to the manufacturer’s instructions. The acquirement was performed using the Navios EX Flow Cytometer (Beckman Coulter, Inc., Brea, CA, USA), and the results were analyzed using Kaluza Analysis Flow Cytometry Software 2.1 and FlowJov10.8.1CL.

### 4.4. Flow Cytometry Antibodies

Flow cytometry antibodies were as follows: Anti-LAG-3-APC (369212, BioLegend, San Diego, CA, USA); anti-PDL1-BV421 (563738, BD Biosciences, Franklin Lakes, NJ, USA); anti-CD200-PE (B68126, Beckman Coulter, Inc., Brea, CA, USA); anti-PD1-PE-Cy7 (329918, BioLegend, San Diego, CA, USA); anti-CD138-PE-CY5 antibody (2109134, Beckman Coulter, Inc., Brea, CA, USA); anti-CD8-PC7 (6607102, Beckman Coulter, Inc., Brea, CA, USA); anti-granzyme B REAfinity™ FITC (130-118-341, Miltenyi Biotec, Bergisch Gladbach, Germany); and anti-Perforin-AF647 (563576, BD Biosciences, Franklin Lakes, NJ, USA).

### 4.5. Sorting of Plasma Cells

Purified PCs were resuspended at the concentration of 1 × 10^6^ cells per 200 µL PBSX1 and stained with anti-LAG-3 antibody. CD138^+^ LAG-3^pos^ or LAG-3^neg^ were sorted with a 100-micron nozzle size, and collected into tubes prefilled with RPMI 1640 medium enriched with 20% FCS (10270106, Gibco^TM^, Thermo Fisher Scientific, Waltham, MA, USA) and 20% Pyruvate (03-042, Biological Industries, Beit-haEmek, Israel). The sorting was done using the BD FACSMelody instrument and BD FACSChorus™ software version 9 (BD Biosciences, Franklin Lakes, NJ, USA). The cells were incubated overnight for recovery.

### 4.6. Co-Culturing of CD138^+^ Plasma Cells with Autologous CD8^+^T Cells

Autologous CD8^+^T cells were cultured alone, or with sorted LAG-3^pos^ or LAG-PCs in the concentration of 2:3 and incubated at 37 °C for 24 h.

### 4.7. Luminex Performance Assay

Supernatants of cultured cells were assessed for IFN-γ; IL-10; IL-17/Il-17A; TNF-α; IL-6; IL-12p70; PD-L1/B7-H1; and VEGF (FCSTM18-08, R&D Systems, Minneapolis, MN, USA), using the Bio-Plex MAGPIX multiplex reader according to the manual’s instructions.

### 4.8. Statistical Analysis

The descriptive statistic was calculated, and a comparison among groups was performed using the Wilcoxon test (for two groups) or the Kruskal–Wallis test, followed by Dunn’s multiple comparisons test (for three groups). The correlations between different clinical and laboratory parameters were identified using the Pearson analysis for multiple correlations and Spearman analysis. The following designations were used in the figures: *: *p* < 0.05, **: *p* < 0.01, ***: *p* < 0.001, ****: *p* < 0.0001 and non-specific: ns. The calculations were made utilizing GraphPad Prism software, version 9.

## 5. Conclusions

LAG-3 expression on malignant plasma cells (PCs) may be involved in the development of MM from MGUS and SMM by decreasing the functional capacity of cytotoxic T cells.

## Figures and Tables

**Figure 1 ijms-25-00549-f001:**
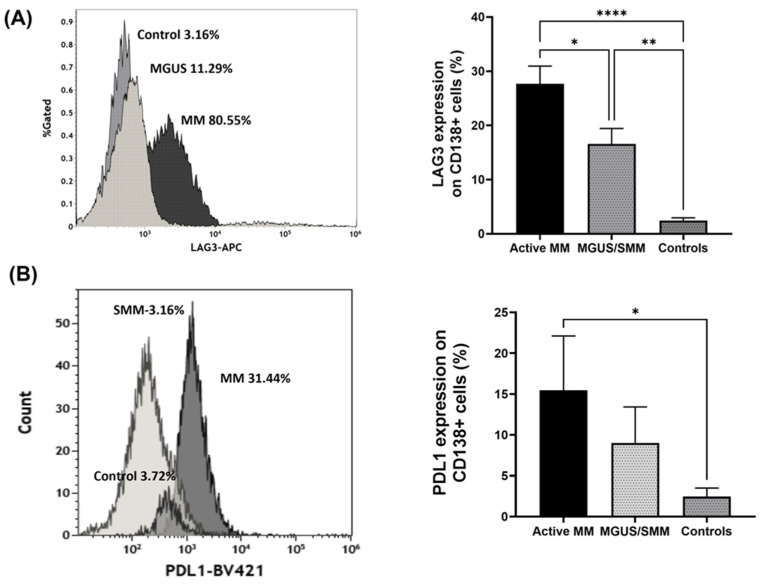
LAG-3 and PDL1 expression on CD138^+^ plasma cells. CD138^+^PCs were purified from fresh bone marrow aspirates from patients with plasma cell dyscrasia. The cells were stained with specific monoclonal antibodies and analyzed by flow cytometry for the expression of LAG-3 and PDL1. (**A**) Representative flow cytometry histograms of LAG-3 expression (%) on purified CD138^+^PCs from MGUS, MM patients vs. control (**left**), with the quantitative analysis of LAG-3 expression (%) on CD138^+^PC (**right**). (**B**) Representative flow cytometry histograms of PDL1 expression (%) on purified CD138^+^PCs from MGUS, MM patients vs. control (**left**); with the quantitative analysis of PDL1 expression (%) on CD138^+^PC (**right**). The following designations were used in the figures: *: *p* < 0.05, **: *p* < 0.01, ****: *p* < 0.0001.

**Figure 2 ijms-25-00549-f002:**
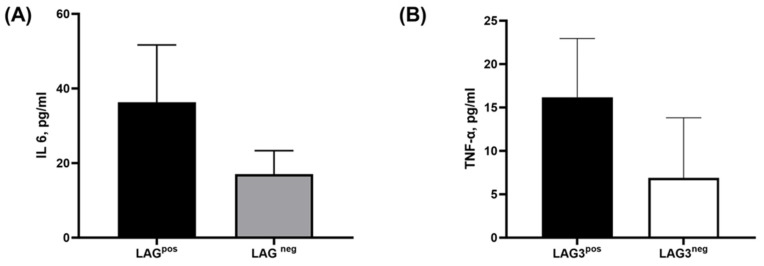
IL-6 and TNF-α secretion from CD138^+^ plasma cells. Purified CD138^+^PCs from bone marrow aspirates (from 8 patients with MGUS/SMM, and 4 patients with active MM), were sorted into LAG-3^pos^ and LAG-3^neg^ and incubated overnight for recovery. On the following day, the supernatants were collected and later analyzed for the secretion of IL-6 and TNF-α by multiplex ELISA. (**A**) The descriptive statistics of IL-6 secretion in supernatants of LAG-3^pos^ vs. LAG-3^neg^ plasma cells. (**B**) The descriptive statistics of TNF-α secretion in supernatants LAG-3^pos^ vs. LAG-3^neg^ plasma cells.

**Figure 3 ijms-25-00549-f003:**
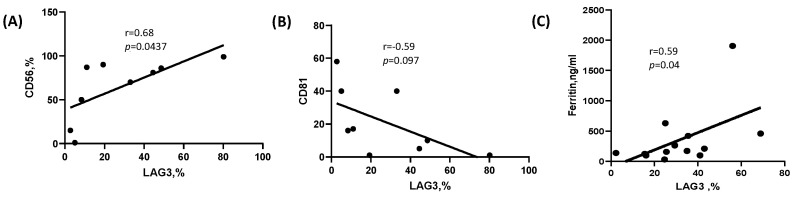
Correlation between LAG-3 expression and diagnostics markers. (**A**) LAG-3 expression (%) on purified CD138^+^ plasma cells, showing correlation with CD56 expression on plasma cells from whole bone marrow evaluated by flow cytometry. (**B**) LAG-3 expression (%) on purified CD138^+^ plasma cells, showing correlation with CD81 expression on plasma cells from whole bone marrow evaluated by flow cytometry. (**C**) LAG-3 expression (%) on purified CD138^+^ plasma cells, showing correlation with ferritin serum levels.

**Figure 4 ijms-25-00549-f004:**
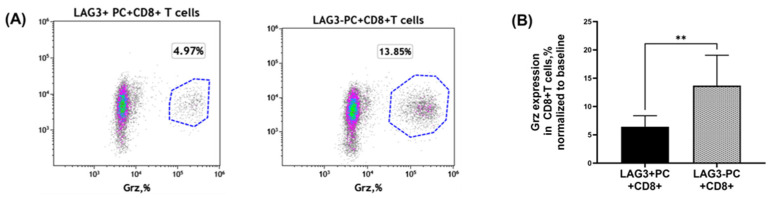
Granzyme expression on CD8^+^T cells. CD138^+^PCs were purified from fresh bone marrow aspirates from patients with plasma cell dyscrasia, and then sorted into LAG3^pos^ and LAG3^neg^ plasma cells. In parallel, autologous CD8^+^ T cells were purified from peripheral blood. On the following day, CD8^+^T cells were co-cultured with LAG3^pos^ or LAG3^neg^ plasma cells at a ratio of 2:3 and incubated for 24 h. On the third day, the cells were stained with antibodies directed against CD8 and granzyme (Grz), and analyzed by flow cytometry. (**A**) Representative flow cytometry density plot of Grz gated on CD8^+^ T cells, co-cultured with either LAG3^pos^ or LAG3^neg^ plasma cells. (**B**) Statistical analysis of Grz expression in CD8^+^ T cells co-cultured with either LAG3^pos^ or LAG3^neg^ plasma cells in 31 patients (17 patients with MGUS/SMM and 14 patients with active MM). Gating strategy after double discrimination: cells, CD8^+^ T cells, Grz on CD8^+^ T cells. The following designation was used in the figure: **: *p* < 0.01.

**Figure 5 ijms-25-00549-f005:**
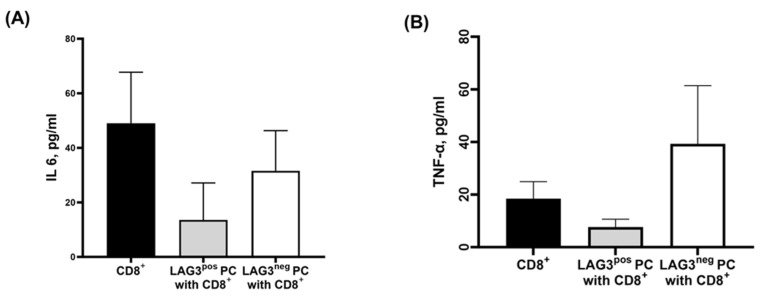
IL-6 and TNF-α secretion from CD8^+^T cells. CD138^+^PCs were purified from fresh bone marrow aspirates from patients with plasma cell dyscrasia and then sorted into LAG3^pos^ and LAG3^neg^ plasma cells. In parallel, autologous CD8^+^ T cells were purified from peripheral blood. On the following day, CD8^+^T cells were co-cultured with LAG3^pos^ or LAG3^neg^ plasma cells at a ratio of 2:3 and incubated for 24 h. The next day, the supernatants were collected and further analyzed for the secretion of IL-6 and TNF- α by multiplex ELISA. (**A**) The descriptive statistics of IL-6 secretion in supernatants of CD8^+^T cells alone, CD8^+^T co-cultured with LAG-3^pos^ vs. CD8^+^T co-cultured with LAG-3^neg^ plasma cells. Krusakal–Wallis test with Dunn’s multiple comparisons test: mean secretion of IL6 by CD8^+^ T cells vs. CD8^+^ with Lag3^pos^PC vs. CD8^+^LAG3^neg^PC is 49.04 vs. 13.58 vs. 31.56 pg/mL *p* = 0.09. (**B**) The descriptive statistics of TNF-α secretion in supernatants of CD8^+^T cells alone, CD8^+^T co-cultured with LAG-3^pos^ vs. CD8^+^T co-cultured with LAG-3^neg^ plasma cells. Kruskal–Wallis test with Dunn’s multiple comparisons test for the mean of TNF-α secretion from CD8^+^ T cells vs. CD8^+^ T cells with LAG3^pos^ PC vs. LAG3^neg^PC: 18.46 vs. 7.64 vs. 39.24, *p* = 0.1.

**Figure 6 ijms-25-00549-f006:**
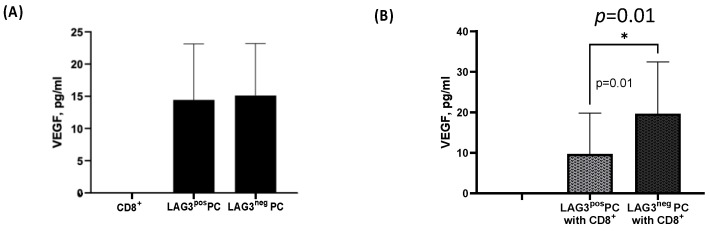
VEGF secretion from CD138^+^ plasma cells and CD8^+^T cells. CD138^+^PCs were purified from fresh bone marrow aspirates from patients with plasma cell dyscrasia and then were sorted into LAG3^pos^ and LAG3^neg^ plasma cells. In parallel, autologous CD8^+^ T cells were purified from peripheral blood. On the following day, CD8^+^T cells were co-cultured with LAG3^pos^ or LAG3^neg^ plasma cells a ratio of 2:3 and incubated for 24 h. The next day, the supernatants were collected and further analyzed for the secretion of VEGF by multiplex ELISA. (**A**) The descriptive statistics of VEGF secretion in supernatants LAG-3^pos^ vs. LAG-3^neg^ plasma cells. (**B**) The descriptive statistics of VEGF secretion in supernatants of CD8^+^T cells alone, and CD8^+^T co-cultured with LAG-3^pos^ vs. CD8^+^T co-cultured with LAG-3^neg^ plasma cells. The following designation was used in the figures: *: *p* = 0.01.

**Table 1 ijms-25-00549-t001:** Clinical and demographic characteristics of patients with MGUS, SMM, NDMM, and RRMM. FISH-high risk was defined as del(17p), translocation t(4;14), translocation t(14;16).

	MGUS n = 17	SMM n = 20	NDMM n = 27	RRMM n = 5
Age, mean ± SD (Year)	69 ± 9	71 ± 10	71 ± 9	63 ± 6
Sex, M/F (n)	10/7	11/9	14/13	2/3
Fish high risk n, %	1	4	6	2
Monoclonal IgG, IgA, IgM, FLC (n)	11,3,2,0	9,6,1,4	12,5,0,6	2,1,0,2
Immunoparesis n, %	3	6	20	5

Immunoparesis is the decrease in one or more uninvolved immunoglobulins below the normal range. MGUS: monoclonal gammopathy of undetermined significance. SMM: smoldering multiple myeloma. NDMM: newly diagnosed multiple myeloma. RRMM: relapsed refractory multiple myeloma.

## Data Availability

The data presented in this study are available on request from the corresponding author.

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
