# Peer review of "The Involvement of LAG-3positive Plasma Cells in the Development of Multiple Myeloma"

_ijms, 2023, doi:10.3390/ijms25010549_

Round 1

Reviewer 1 Report

Comments and Suggestions for Authors

The LAG-3 inhibitory receptor is expressed on regulatory plasma cells (PCs). Micro-environmental cells expressed LAG-3 were found to be increased during the progression of smoldering multiple myeloma. While most studies are focused on the LAG-3 role in the microenvironment, less is known about LAG-3 expression on malignant PCs and its possible role in the progression of MM from MGUS and SMM. A growing interest in LAG-3 was influenced by the recent discovery of a new subgroup of PCs in mice, called regulatory PCs, which express LAG-3 and secrete IL-10. These cells have not been studied in humans, and there is no data about their role in the pathogenesis of MM.  Herein, the authors assess the possible role of LAG-3 expression on regulatory PCs in patients with plasma cell dyscrasia and considered that malignant PCs expressing LAG-3 are actively involved in the pathogenesis of MM progression from MGUS and SMM by impairing the function of effector immune cells. This is an interesting story. But I have several following concerns:

1. Abbreviations should be defined when they first appear in the text. Such as “LAG3”, "MM", “CD138”...

2. Tables should use a standard three-line table and should not span pages.

3. Please add figure ligends for all the Figures in this manuscript.

4. "p" representing significant differences should be italicized.

5. Please add the abscission and gate method of the flow scatter plot in Figure 4.

6. Please make a statistical analysis of the results in Figure 5.

7. Please also add a description of human ethics to the text.

8. Please unify the format of references in the article, including the author's name, the case of words in the title of the article, the writing of the name of the journal, and the page number.

Comments on the Quality of English Language

Minor editing of English language required.

Author Response

Dear Reviewer 1,

First of all, we would like to thank you for your constructing comments.

We would like to answer, point by point, to these comments as follow:

  1. We defined all the abbreviation as was requested. See all the definitions marked in yellow.
  2. We changed the Table, as was requested. See Table 1, lines 111-113.
  3. We added the figure legends for all figures. Please see the new manuscript.
  4. We changed the "p" to be italicized in the manuscript as requested.
  5. We added the gate strategy as requested. Please see line 201 marked in yellow.
  6. We added the statistical analysis for figure 5. Please see lines 224-229 marked in yellow.
  7. We added more details on the human study ethics approval. Please see lines 348-350 marked in yellow.
  8. We unified the format of the references to match the journal requirement.

Reviewer 2 Report

Comments and Suggestions for Authors

The manuscript entitled: “The involvement of LAG-3pos plasma cells in the development of

multiple myeloma” (ijms-2760031) by Kreiniz et al. aims to evaluate the role of LAG-3 expression on regulatory plasma cells in patients with plasma cell dyscrasia.

Albeit the paper is well written, prepared and of special interest, comments should be addressed to further improve the manuscript.

Comments:

1.    The authors should explain the reasons for using patients with normal bone marrow biopsie as controls.

2.    Page 3, line 120: “premalignant condition”: the authors should define this more clearly at this point.

3.    Page 4, line 139-140: These results should be also more discussed within the discussion section.

4.    Discussion section section: the authors should more deeply discuss their results within this section, since their current results are not confirmed and additionally the discussion section should be balanced according new insights (which should be more highlighted) and limitations of their study.

5.    Page 3 line 104: please use percentage within the whole manuscript where appropriate.

Author Response

Dear Reviewer 2,

First of all, we would like to thank you for your constructing comments.

We would like to answer, point by point, to these comments as follow:

  1. We used control bone marrow sample in order to described and compare normal plasma cells to the diseased plasma cells. This comparison is important and needed for two reasons; first, to define and describe actual regulatory plasma cells in human bone marrow. The second reason is to describe the evolution of these regulatory plasma cells from healthy to malignant cells.
  2. We defined the premalignant conditiones: MGUS and smoldering MM patients, see lines 122 marked in yellow.
  3. We added to the "Discussion" section a paragraph dealing with PDL1 and CD200 results as was requested. Please see lines 277-291 marked in yellow.
  4. We added to the "Discussion" section dealing with the limitation of the study. Please see lines 339-342 marked in yellow.
  5. We added the percentage as requested. Please see lines 100-110 marked in yellow.

Round 2

Reviewer 1 Report

Comments and Suggestions for Authors

The authors have addressed all my concerns. I recommend accepting it in current form.

Reviewer 2 Report

Comments and Suggestions for Authors

The manuscript entitled: “The involvement of LAG-3pos plasma cells in the development of

multiple myeloma” (ijms-2760031) by Kreiniz et al. aims to evaluate the role of LAG-3 expression on regulatory plasma cells in patients with plasma cell dyscrasia.

After revision of the manuscript, the authors addressed all my initial comments sufficiently.